# New Technique for Enhancing Residual Oil Recovery from Low-Permeability Reservoirs: The Cooperation of Petroleum Hydrocarbon-Degrading Bacteria and SiO_2_ Nanoparticles

**DOI:** 10.3390/microorganisms10112104

**Published:** 2022-10-24

**Authors:** Kai Cui, Hailan Li, Ping Chen, Yong Li, Wenxue Jiang, Kun Guo

**Affiliations:** 1School of Chemical Engineering and Technology, Xi’an Jiaotong University, Xi’an 710049, China; 2State Key Laboratory of Heavy Oil Processing, China University of Petroleum, Beijing 102249, China; 3NCO Academy, Space Engineering University, Beijing 101416, China; 4Drilling and Production Engineering Technology Research Institute of CNPC Chuanqing Drilling Engineering Co., Ltd., Xi’an 710018, China; 5National Engineering Laboratory for Exploration and Development of Low Permeability Oil and Gas Fields, Xi’an 710018, China

**Keywords:** petroleum hydrocarbon-degrading bacteria, silica nanoparticles, residual oil mobilization, visual micromodels, microbial-nanofluids, enhance oil recovery

## Abstract

Residual crude oil production from low-permeability reservoirs has become a huge challenge because conventional recovery techniques are inefficient. Nanofluids as a new type of oil-displacement agent have become a hot topic in recent years to enhance oil recovery (EOR) in reservoirs. However, the imperfection of agglomeration, dissolution, and instability of nanofluids in reservoir environments limit their ability to drive oil. Here, a novel “microbial-nanofluid” composed of petroleum hydrocarbon-degrading bacteria (PHDB, namely *Bacillus cereus*) and SiO_2_ nanoparticles was proposed as a potential new technology for enhancing residual oil recovery. The micromodel displacement test results showed that the optimum composite concentration of “microbial-nanofluids” were PHDB (7.0%, *v*/*v*) and SiO_2_ nanoparticles (100 mg/L), and the residual oil recovery was increased by 30.1% compared with waterflooding (68.8%). Moreover, the morphological characteristics of residual oil mobilization after “microbial-nanofluid” flooding were mainly small and dispersed oil droplets in the excessive areas, and the dead-end areas were almost clean with mobilization. Furthermore, the cooperation mechanism of four kinds of “microbial-nanofluids” to enhance the residual oil recovery in low-permeability reservoirs was preliminarily clarified, namely the co-emulsification of oil, working together to unclog oil clog, microbial-nanofluid self-assembly, and structural disjoining pressure. This study demonstrated that PHDB-SiO_2_ nanoparticle composite flooding technology provided a significant potential for the EOR from low-permeability reservoirs.

## 1. Introduction

Residual crude oil production from low-permeability oil fields has become a huge challenge because conventional recovery techniques are inefficient. At present, advanced water flooding techniques, in combination with optimized good pattern arrangement, fracturing, chemical, and microbial enhanced oil recovery (MEOR) methods, are the main exploitation techniques for residual crude oil from reservoirs [1,2]. However, these conventional recovery techniques can only produce less than 40–50% of the total crude oil reserves, and there is a large amount of residual oil in the reservoir that cannot be dislodged [3]. Meanwhile, the oil-wet and low-permeability properties of low-permeability reservoirs reduce the oil-displacement efficiency and sweep efficiency of injected fluids [4]. When the rock properties are heterogeneous, the displacing fluid mostly flows through the high permeability zone, while the low-permeability zone remains unswept. According to the introduction of the oil recovery process [2,5], the formation mechanism, volume, and spatial distribution of residual oil are important research contents for the oil field reservoirs evaluation in different development stages and enhanced oil recovery (EOR). Thus, the difficulty to enhance residual oil recovery from low-permeability reservoirs requires the development of an alternative and cost-effective crude oil recovery process.

Compared with traditional tertiary oil recovery technologies, nanofluids as a new type of oil-displacement agent have become a hot topic in recent years to enhance oil recovery in low-permeability reservoirs, where silica (SiO_2_)-based nanoparticles have been most commonly used [6,7]. The nanoparticles exhibit marvelous interfacial behavior due to their nanoscale size and large specific surface area [8]. Hendraningrat et al. utilized hydrophilic SiO_2_ nanoparticles to prepare nanofluids and studied the effect of nanofluid flooding on enhanced oil recovery of low-medium permeability sandstone [9]. In addition, the co-adsorption of nanoparticles and surfactants on oil/water or oil/water/solid can affect the interface energy, resulting in interfacial tension reduction and wettability alteration [10,11]. Nwidee et al. revealed that nanoparticle–biosurfactant systems exhibited an excellent ability as a new EOR agent under reservoir conditions [12].

Nanofluid is defined as a stable and homogeneous suspension obtained by adding nanoparticles with a 1–100 nm average size in a traditional liquid medium such as water, oil phase, or alcohol [7,11]. Even though nanofluids have shown great potential in the EOR process, the nano-suspension prepared by pure nanoparticles or nanoparticles and biosurfactants cannot obtain effective production. Meanwhile, these nanofluid products show good oil-displacement performance in the laboratory environment, but they often have agglomeration, dissolution, and instability defects in the real reservoir environment [13], which limit their oil-displacement ability. For example, surfactants are adsorbed and retained in the reservoir, polymers degrade under high temperatures and high salinity, alkali flooding and combination flooding corrode pipelines in the process of oil production, and scale is easy to appear in the reservoir [10,11]. Based on these challenges, it is necessary to modify the action environment of SiO_2_ nanoparticles to achieve the flooding effect of the composite system.

MEOR involves the use of indigenous or specially screened microorganisms to produce specific metabolites by injecting the formation or ground fermentation to enhance crude oil recovery. In MEOR, indigenous microbial-enhanced oil recovery technology has been recognized to have the advantages of strong adaptability, good compatibility with the reservoir, and low cost [14]. Reservoir microorganisms can facilitate the mobilization of oil through the production of amphiphilic compounds, termed biosurfactants, which reduce the interfacial tension between immiscible phases. In microbial flooding, crude oil is degraded by the synergistic effects of microbes and metabolites, leading to improved swept volume and EOR [15]. In particular, petroleum hydrocarbon-degrading bacteria (PHDB) are the major participants in the start-up stage of microbial flooding because they can utilize petroleum hydrocarbons as the sole carbon source [16]. A great quantity of PHDB such as *Pseudomonas aeruginosa* and *Bacillus subtilis* may survive under high-temperature, high-pressure, and oligotrophic oil reservoir conditions [17]. Moreover, the PHDB degrades n-alkanes and cycloalkanes with different carbon numbers without other carbon sources [18]. Since the biosurfactants from the metabolization by PHDB stimulate crude oil degradation by enhancing bioavailability [19], it is worth noting that the PHDB growth consumes very little crude oil relative to reservoir reserves, so it does not affect the quality of crude oil. On the contrary, according to the lab’s previous research [2], PHDB can degrade the heavy components of crude oil (such as aromatic compounds (phenanthrenes, benzenes, naphthalenes, and thiophenes)) and make it lighter to improve the quality of the oil. Although many studies have explored the single or synergistic relationship between SiO_2_ nanoparticles and biosurfactants for EOR [7,20], few works have reported on the synergistic mechanisms of PHDB with SiO_2_ nanoparticles to EOR, and there is a serious lack of data to visually characterize the distribution and migration of the residual crude oil with the synergistic effect of PHDB and SiO_2_ nanoparticles.

Based on the above considerations, provided that the advantages of adaptability and surfactant production of PHDB can be used to provide the biosurfactant with SiO_2_ nanoparticles in the reservoir to construct a “microbial-nanofluid”, which is to make up for the imperfection of bio-nanofluids, the purpose of this study was to analyze and reveal the distribution and migration characteristics of residual crude oil with the single or synergistic effect of PHDB and SiO_2_ nanoparticles through visual glass micromodel tests, and its mechanism for EOR. The micromodel experiments were designed to present the visualization process of different displacement fluids (production water, SiO_2_ nanoparticles, PHDB, and PHDB-SiO_2_ nanoparticles) for residual oil, and investigate the effects of parameters (injection rate, fluids concentration, and fluids composition) for the residual oil ratio and EOR. Furthermore, heterogeneous micromodels generated based on CT images of real rocks were used to evaluate the EOR process in porous media that is more representative of reservoirs. This work proved the potential of PHDB-SiO_2_ nanoparticle composite flooding for EOR and showed how the “microbial-nanofluids” affected the residual oil mobilization in low-permeability reservoirs.

## 2. Materials and Methods

### 2.1. Materials

The crude oil utilized in the experiments was collected from a low-permeability reservoir located in the Xinjiang oilfield of China National Petroleum Corp. The density of the oil was 0.879 g/cm^3^, and its viscosity was 18.2 mPa·s at 30 °C. The SiO_2_ nanoparticles with a diameter of 10–20 nm and a purity of 99.5% used in this study were purchased from Aladdin Reagent Co., Ltd. The specific surface area and density of the SiO_2_ nanoparticles were 180–270 m^2^/g and 2.0–2.5 g/cm^3^, respectively. Additionally, the flooding water used in the experiments was the produced water collected from an oil well in the Xinjiang oil field, China. The properties of the produced water are presented in the Appendix A.

The PHDB strain (Appendix A) used in this experiment was obtained from the production fluid of the Xinjiang oil field in China, which exhibits a similarity of 98–99% to *Bacillus cereus* according to the 16S rRNA gene analysis. The experimental process of the PHDB isolation is as follows: 10 mL of the produced water and 90 mL of PHDB screening medium (molasses 1.47 g/L, NaNO_3_ 5.33 g/L, (NH_4_)_2_SO_4_ 2.67 g/L, NaCl 2.0 g/L, KH_2_PO_4_ 10.0 g/L, Na_2_HPO_4_ 4.0 g/L, MgSO_4_ 0.5 g/L, and yeast powder 0.6 g/L)) were mixed at 45 °C with shaking at 150 r/min for 3–5 days. Then, 5% of the mixed bacteria liquid was inoculated into 100 mL of fresh screening medium. The tubes were incubated for 3–5 days under facultative anaerobic conditions at 45 °C with shaking at 150 r/min. We repeated the above steps several times, and it was then stretched on the Petri dish to pick a single PHDB colony.

### 2.2. Preparation of the Displacement Fluids

Water displacement fluid: The produced water was sterilized to remove microorganisms and then used as the water displacement fluid.

PHDB displacement fluids: To activate the preserved PHDB strain, 5.0 mL of the PHDB preservation solution was added to 100 mL of LB medium (peptone 10 g/L, yeast powder 10 g/L, and NaCl 10 g/L) for cultivation at 45 °C for 24 h. Then, 5.0 mL of the activated PHDB was inoculated into 95 mL of the screening medium for enrichment at 45 °C. When the concentration of PHDB reached 10^5^–10^6^ cells/mL, the bacteria solution was added to sterilized produced water (3.0%, 5.0%, and 7.0%, *v*/*v*) to prepare the PHDB displacement fluids.

SiO_2_ nanoparticle displacement fluids: The SiO_2_ nanoparticles were added to sterilized produced water (50 mg/L, 100 mg/L, and 200 mg/L) to prepare the SiO_2_ nanoparticles’ displacement fluid. Before the displacement experiment, the SiO_2_ nanoparticle solution was mixed for 15 min in an ultrasonic crusher (TL-T650CT, Xiangtan Xiangyi Instrument Co., Ltd, Hunan China) to ensure uniform suspension.

PHDB-SiO_2_ nanoparticle composite displacement fluid: Based on the results of previous laboratory studies [5,7,19], the preparation standards of PHDB-SiO_2_ nanoparticle composite displacement fluid were designed. The SiO_2_ nanoparticles (100 mg/L) were added to the PHDB displacement fluid (7.0% of PHDB) to prepare the PHDB-SiO_2_ nanoparticle composite displacement fluid.

### 2.3. Preparation of Glass Micromodels

According to the method designed by Richard et al., the laser etching technique was used to fabricate glass micro-etching models [21]. Glass micromodels are quasi-2D representations of porous media that allow direct visualization of pore-scale fluid flow [7]. The micromodels consist of two glass plates: One is an etched plate, with another fine cover glass plate annealed over the etched pattern on the facing glass plate in an oven with a temperature cycle reaching a maximum of 700 °C. In this study, the glass micromodels used in the experiment are a transparent simulation model with pore-throat characteristics close to the natural core, which is made by laser etching technology based on the core slices of low-permeability reservoirs in Xinjiang, as shown in Appendix A. The production of synthetic micromodel samples was commissioned by the China University of Petroleum (Beijing), and the preparation process was kept confidential. The length, width, and thickness of the micromodel are 75, 75, and 3 mm, respectively, the average effect size is 65 × 65 mm, and the average pore radius of the obtained template is 20–40 μm. In addition, the micromodels were washed with dichloromethane and deionized water during the preparation process and burned in a muffle furnace at 500 °C for 2 h to remove organic residues followed by cooling to ambient temperature.

### 2.4. Micromodel Displacement Experiments

The schematic diagram of the displacement experimental equipment process is illustrated in Figure 1. The oil–water area, distribution, and morphology of residual oil mobilization under the different displacement fluid conditions were investigated by the visual micromodel experiments, and the oil recovery was calculated (Equation (1)). This system consisted of an inverted microscope for observing the distribution of residual oil in the micromodel before and after different fluids’ displacement. A syringe pump was used to inject fluids at low rates. A camera was placed above the micromodel for capturing images at defined time intervals. During the whole process of displacement, the experimental temperature was controlled at 45 °C. The experimental design is shown in Table 1.

A flow chart of displacement experimental procedures in the micromodels is shown in Appendix A. The specific displacement procedures were as follows: (1) Sterilized crude oil was pumped into the vacuumed micromodel till it reached oil saturation. (2) For I primary water flooding, the tested injection rates of the produced water were 0.05, 0.25, and 0.5 mL/min, and the optimal primary water flooding rate (to simulate the residual ratio of crude oil in the real reservoir) was obtained. (3) After primary water flooding (0.05 mL/min), the PHDB displacement fluids (3.0%, 5.0%, and 7.0%, respectively) were injected into the micromodel by the alternating slug method with in injection volume of 3.0 pore volume (PV), and then, the ends of the model were closed and incubated at 45 °C for 5 days. For the experimental control group, the flooding fluid was the produced water after sterilization. (4) When the incubation ended, secondary water flooding (0.05 mL/min) was resumed immediately until the water content was greater than 98% or the accumulation of flooding reached 5.0 PV. The experiment was then stopped. When the displacement tests ended, the volumes of produced oil and water were measured by the calibration oil–water separator, and finally, the images of the distribution morphology of residual oil mobilization were obtained. The displacement procedures of SiO_2_ nanoparticles fluids (50, 100, and 200 mg/L, respectively) and PHDB-SiO_2_ nanoparticle composite displacement fluid were the same as that of PHDB displacement. The oil recovery efficiency ($\eta_{1}$, %) was calculated by the following formulas:(1)$$\eta_{1}=(V_{d}/V_{0})\times 100\%$$
where $V_{d}$ (mL) was the cumulative oil output after displacement; and $V_{0}$ (mL) was the saturated oil in the model before displacement.

### 2.5. Analysis of Residual Oil

Calculation of residual oil ratio: The images obtained from the displacement experiments were analyzed to obtain the distribution characteristics and formation of the residual oil under different displacement fluids, and the residual oil ratios were calculated. Image Magick software was used to calculate the proportion of a certain color in the picture. The residual ratio was calculated by the change of the crude oil color ratio before and after displacement. The specific calculation method is as follows: (1) Cut the calculated images to reduce interference and improve the accuracy of the calculated data; (2) output the color information of each pixel in the picture to generate a TXT file, with each line containing one pixel; (3) screen the rows that meet the requirements to attain the pixels that meet the requirements; (4) by analogy, as many lines as there are screened out, there are as many pixels that meet the requirements; (5) calculate the area proportion of red (Equation (2)):Residual oil ratio (%) = (original oil color area–remaining oil color area)/original oil area × 100%(2)

Determination of size and size distribution of emulsified residual oil: The image of emulsified crude oil was observed by a microscope (Olympus BX51, Tokyo, Japan) at room temperature. The droplet size and size distribution were obtained by statistical analysis using the software provided by the microscope supplier.

## 3. Results and Discussion

### 3.1. Migration Characteristics of Crude Oil under Primary Water Flooding

To simulate the distribution characteristics of residual oil migration in low-permeability reservoirs, the distribution and residual ratios of residual oil under different water flooding rates (0.05, 0.25, and 0.5 mL/min) were investigated and analyzed (Figure 2a–c), and the corresponding oil recovery was calculated, respectively (Figure 2d). When the water flooding rates were 0.05, 0.25, and 0.5 mL/min, the residual ratios of crude oil were 36.8%, 18.4%, and 14.5%, and the average recovery efficiencies were 68.8%, 83.5%, and 84.5%, respectively. These results implied that the crude oil recovery efficiency increased with the increase in injection rates, but the increasing trend slowed down when it reached a certain injection rate. The primary water flooding can only sweep the crude oil in the middle channel and the excessive region due to the physical properties of the fluid, and the boundary blind end crude oil cannot be driven out. In general, about 50–60% of the residual crude oil in the reservoir after water flooding [4]. Thus, 0.05 mL/min was selected as the oil-displacement rate to carry out the micromodels in the subsequent experiments, because the residual oil distribution at this time was closer to the actual residual oil ratio of the reservoir after water flooding.

In addition, the distribution and morphology of residual oil migration at the water flooding rate of 0.05 mL/min were observed (Appendix A), and it was found that the distribution characteristics of residual oil were mainly manifested in the following six forms: (1) the residual oil in the oil film form (Appendix A); (2) the residual oil in the cluster form (Appendix A); (3) the residual oil in the oil droplet form (Appendix A); (4) the residual oil in the solitary island form (Appendix A); (5) the residual oil in the columnar form (Appendix A); and (6) the residual oil in the blind end form (Appendix A). These characteristics occurred with the continuous water displacement process, due to the low viscosity and single property of water, which would be difficult to affect some small pores of residual oil and dead-end oil [22], resulting in a large amount of crude oil being stuck in the reservoir, and ultimately reducing the oil recovery.

### 3.2. Effects of Displacement Fluids on Residual Oil Recovery

The saturation and recovery efficiency of residual oil were measured under different displacement fluid (SiO_2_ nanoparticles and PHDB) conditions to determine the composite concentration of PHDB-SiO_2_ nanoparticle displacement fluids (Table 2). For SiO_2_ nanoparticle displacement, when the SiO_2_ nanoparticle concentrations increased from 50 to 200 mg/L, the residual oil saturation first decreased and then increased with the increase in concentrations, and the lowest saturation value at 100 mg/L was 4.8%. The maximum recovery efficiency was 86.5%, which was higher than that of 50 (82.6%) and 200 mg/L (84.4%). These phenomena demonstrated that too high a concentration of SiO_2_ nanoparticles does not necessarily favor the stripping of crude oil caused nanoparticle emulsion. In contrast, when the SiO_2_ nanoparticles concentration was 100 mg/L, this prevented the excess aggregation of nanoparticles in an ultra-low-permeability core [5,7]. In addition, it also can be observed from Figure 3a that the morphology of residual oil mobilization under SiO_2_ nanoparticle displacement (100 mg/L) was mainly cluster, solitary island, and oil film, while the residual oil in the oil droplet form was less. These phenomena may be attributed to the differential pressure dramatically increasing because of the adsorption and deposition of SiO_2_ nanoparticles in pore spaces [23]. Based on this pressure, the flaky oil was broken, and then it dredged the porosity channels trapped with residual oil.

In PHDB displacement, the recovery efficiency increased, and residual oil saturation decreased continually with increasing the concentrations of PHDB bacteria solution (Table 2). When the PHDB bacteria solution concentration was 7.0%, the maximum recovery efficiency and lowest residual oil saturation were 90.5% and 2.1%, respectively. This is because a high concentration of PHDB does necessarily favor the emulsion of crude oil caused by biosurfactants formed from biodegradation, and ultimately contributing to enhancing oil recovery [3,19]. In addition, compared to water displacement and SiO_2_ nanoparticle displacement, the formation and quantity of columnar residual oil and oil droplet residual oil increased significantly under PHDB displacement (7.0%) (Figure 3b), indicating that the emulsification of crude oil greatly improved the dispersion of oil, thereby increasing the contact probability between microbes and oil droplets. During this process, the PHDB themselves could break the oil into small droplets [24], which accelerated the formation of oil in water and dramatically increased the oil recovery (more than 90%). In addition, we found that the small oil droplets dispersed in the channel pore-throats were aggregated into large oil droplets under the action of PHDB, which was more conducive to the recovery of residual oil. Considering the factors of recovery efficiency and synthesis cost, both 7.0% of PHDB and 100 mg/L of SiO_2_ nanoparticles were determined as the optimal composite concentration of the PHDB-SiO_2_ nanoparticle displacement fluids.

The distribution characteristics and formation of residual oil mobilization under the optimal composite concentration of PHDB-SiO_2_ nanoparticle displacement fluids were further observed and investigated (Figure 3c). After PHDB-SiO_2_ nanoparticle composite displacement, the saturation and recovery efficiency of residual oil reached 0.8% and 98.9%, respectively, which would be significantly better than a single PHDB or SiO_2_ nanoparticle displacement. Compared with secondary water flooding (68.8%), PHDB-SiO_2_ nanoparticle composite flooding enhanced the recovery of residual oil by 30.1%. This is because the synergistic action of the PHDB and SiO_2_ nanoparticles further split smaller oil droplets (Figure 3c), reducing their flow resistance and enhancing oil recovery. On the other hand, the addition of SiO_2_ nanoparticles into the PHDB bacteria solution promoted the ordering of nanoparticles in the wedge film region due to the in situ production of biosurfactant by PHDB [19,25], thus increasing g the dredging of the pore-throat residual oil and enhancing the oil-displacement efficiency. At present, the mechanism of how PHDB use bacteria or metabolites to modify SiO_2_ nanoparticles to improve their dispersion stability is not clear.

One of the most commonly used methods for evaluating the dispersion stability of the emulsion is by microscopic observation of the droplet size distribution of crude oil. According to the basic theory of emulsion [26], the smaller the droplet size and the more uniform its distribution, the more stable the emulsion. Therefore, to further prove the emulsification and dispersion ability of the PHDB-SiO_2_ nanoparticle composite displacement on the residual oil, the effects of different displacement fluids on the droplet size of residual crude oil were investigated (Figure 3d). Crude oil was not well emulsified before displacement but dispersed as big droplets in the production water. The droplet size distribution shows that most of the oil droplet sizes were larger than 15 μm (counted to 61%). After different displacement fluid actions, most of the residual oil droplets had sizes between 2 and 15 μm. The residual oil droplets with sizes of 2–15 μm averaged 64% in the SiO_2_ nanoparticle displacement, which increased to 84% in the PHDB displacement and then gradually increased to 90% in the PHDB-SiO_2_ nanoparticle composite displacement. These results indicated that the PHDB-SiO_2_ nanoparticle composite displacement formed a synergistic effect favoring an enhanced emulsion of the crude oil. The emulsification of residual oil greatly improved the dispersion of oil. Therefore, the finely dispersed crude oil droplets were able to be displaced from the pores of the oil reservoir so that oil recovery was enhanced. Additionally, considering the abundance of PHDB in the real reservoir environment, the industrial PHDB-SiO_2_ nanoparticle composite displacement fluid is mainly microbial nutrient activators and SiO_2_ nanomaterials. The nutrient activates PHDB in situ in the reservoir while coupling with the SiO_2_ nanoparticles to emulsify and displace the residual crude oil, and to EOR.

### 3.3. Mobilization Characteristics of Residual Oil in Different Displacement Areas

To further explain the EOR caused by PHDB-SiO_2_ nanoparticle composite displacement, the formation characteristics of residual oil mobilization in different displacement areas were studied in the micromodels (Appendix A). According to the literature [27], the model (Appendix A) is mainly divided into three regions: middle channel areas, excessive areas, and boundary dead-end areas.

#### 3.3.1. Middle Channel Areas

After water flooding, a small amount of the residual oil existed in the middle channel areas in the form of the states of the droplets and clusters (Figure 4a), which indicated that most of the residual oil in this area was displaced because of the effective sweep of water injection. Compared with water displacement, Figure 4b,c show that there is almost no remaining oil in the channel pore of the middle areas, indicating that the PHDB or SiO_2_ nanoparticles have a good displacement effect on the residual oil in the middle channel pore. In Figure 4d, the residual oil was almost completely displaced in middle channel areas after PHDB-SiO_2_ nanoparticles composite displacement because the PHDB itself and its metabolites could split the large oil droplets into smaller ones that could pass through the water flooding channels with the help of nanoparticles.

#### 3.3.2. Excessive Areas

Figure 5 compares the mobilization characteristics of the residual oil on the upper and lower sides of the model position (namely, excessive areas) for different displacement fluids, which would show the movement of the residual oil through the channel pores. Compared with the middle channel position, the residual oil ratio in excessive areas was higher than that in the middle channel areas because the water displacement swept volume in this area was smaller than that in the middle position (Appendix A). During the water displacement process, the residual oil in the excessive areas mainly existed in the form of solitary island oil and columnar oil in the channel pores, and the amount of solitary island oil was the majority (Figure 5a). In Figure 5b, the residual oil in an excessive position was not displaced clean by SiO_2_ nanoparticle displacement, but the solitary island state of residual oil in this area was split into small columnar oil droplets, which improved the fluidity of the residual oil [5,28]. Nevertheless, as depicted in Figure 5c, the columnar residual oil was emulsified into smaller particle-size oil droplets under the action of a high biomass concentration of PHDB and was partially displaced, indicating that the microorganisms combined with its biosurfactant effectively improve sweep volume. This may be attributed to the biodegradation and biosurfactant formation caused by the reduction in surface tension of the oil–water and the emulsification of crude oil in the aqueous phase [29]. Meanwhile, when the PHDB coordinated with SiO_2_ nanoparticles for further displacement (Figure 5d), it was obvious that all the residual oil was emulsified and split into oil droplets with small particle sizes so that the oil film attached to the surface of the pore was also stripped off, which effectively increased the emulsification effect and fluidity of the crude oil, and further enhanced oil recovery. Therefore, reservoir sweep volume is an important issue to be considered in EOR, regardless of the types of oil recovery displacement fluids in the future.

#### 3.3.3. Boundary Dead-End Areas

Another reason for the difficulty of the EOR from low-permeability reservoirs is the existence of many redundant dead-end pores [7,30]. In the close-up images of the boundary dead-end in the micromodel test, as shown in Figure 6, the interaction between the different displacement fluids and residual oil in the dead-end pores was further analyzed. Some of the dead-end oil would be displaced after water flooding, and the residual oil mainly existed in the form of clusters and solitary islands in the channel pores (Figure 6a). These phenomena illustrated that the dead-end pores remained full of residual oil after water displacement, exhibiting poor treatment impact for displacing oil from such pores. The swept volume was small because the displacing fluid could only be in contact with one side of the oil in dead-end pores [31]. Additionally, we found that when PHDB collaborated with SiO_2_ nanoparticles were used for residual oil displacement, the blind end crude oil was more likely to be divided into smaller particle sizes (Figure 6d), and the oil-displacement efficiency was much higher than that of single PHDB or SiO_2_ nanoparticles, respectively (Figure 6b,c). These experimental phenomena showed that the wettability of dead-end pores translated to a water-wet state avoiding the secondary accumulation of motivated oil after PHDB-SiO_2_ nanoparticle composite flooding. Further, the oil entrapped in the small pores or dead-end pores was displaced more efficiently by the displacement of the synergy with PHDB-SiO_2_ nanoparticles than the PHDB or SiO_2_ nanoparticles, and the PHDB displacement efficiency was second. Because of their small size, nanoparticles can enter into the pore space of dead-end areas with the help of PHDB or biosurfactant, not only “scraping” residual oil from small pores but also dredging porosity channels to increase the sweep efficiency of the water flooding.

### 3.4. Theoretical Mechanism Analysis of Microbial-Nanofluids for EOR

The various distributions of residual oil in channel pore spaces in the low permeability for the small pore size and complex strata structure of the reservoir, and the corresponding oil-displacement mechanism is also different. Based on the above research results of visual microscopic glass model experiments, the synergistic mechanism of four kinds (Paths 1–4) of PHDB-SiO_2_ nanoparticle composite flooding to enhance the residual oil recovery in low-permeability reservoirs was preliminarily clarified, as illustrated in Figure 7.

Figure 7c explained the process of dredging the oil-clogged pore-throats with the PHDB-SiO_2_ nanoparticle composite flooding. The PHDB degrades n-alkanes and cycloalkanes with different carbon numbers without other carbon sources [33]. In this growth process, the residual oil is gnawed and degraded by synergistic effects of PHDB and metabolites, resulting in the adsorption and deposition of nanoparticles in pore gap spaces. Because of the small size, SiO_2_ nanoparticles can enter the pore space of low-permeability reservoirs, not only “scraping” residual oil from small pores but also dredging porosity channels to increase the sweep efficiency of the water flooding (Path 2). The pore-throat with trapped oil was dredged which increased the residual oil recovery efficiency. Additionally, PHDB and its biosurfactants can effectively form synergistic action to emulsify and disperse the residual oil into smaller oil droplets, thus increasing the migration and deformation capacity of the residual oil and further promoting the flow of the residual oil in the channel pores.

The dead-end pores remained full of residual oil after water flooding (Figure 6a), exhibiting poor treatment impact for displacing oil from such pore spaces. In PHDB-SiO_2_ nanoparticle composite flooding, PHDB acted as transporters to transport and stabilize the SiO_2_ nanoparticles at the blind end positions. The water path near the dead-end pore was further expanded, and the trapped oil was “dug out” from the dead-end pore (Figure 6d and Figure 7d). Thus, it demonstrated that the SiO_2_ nanoparticles can be attached to the rock surface or on the surface of the oil layer, further changing the wettability on the surface of the rock or reducing the oil–water interfacial tension [7], prompting the biosurfactant and SiO_2_ nanoparticles to parcel crude oil to form a water-in-oil emulsion, and eventually led to the crude oil being stripped and displaced from the rockface or dead-end side (Path 3). On the other hand, the Brownian motion increased the contact of nanoparticles with residual oil, which is propitious to motivate trapped oil in dead-end pores [34].

In PHDB-SiO_2_ nanoparticle composite displacement, the residual oil was stripped off from rock surfaces by the structural disjoining pressure (Figure 7e). Based on the theory of structural disjoining pressure [35,36], SiO_2_ nanoparticles converge on a three-phase contact region (oil drop/nanofluid/rock surface) and form a wedge film. The ordering of the nanoparticles in the wedge film region results in excess pressure termed structural disjoining pressure. The direction of the pressure is toward the vertex of the wedge film region. When SiO_2_ nanoparticles were dispersed in the PHDB bacteria solution, the situ production of biosurfactants was absorbed onto the particle surface via electrostatic attraction and formed a new monomer layer. The monomer layer can not only promote the forward migration of the wedge film region, but also weaken the adsorption of residual oil on the rock surface and reduce the difficulty of stripping oil with nanoparticles. Moreover, the residual oil was easily stripped from pore wall surfaces because the self-assembly of biosurfactants and SiO_2_ nanoparticles promotes the orderly distribution of nanoparticles in the wedge film region. In addition, the “water film” (the self-assembly production of biosurfactants and SiO_2_ nanoparticles) that formed between the rock surface and oil film differed from that in the waterflooding and SiO_2_ nanoparticles flooding treatment [37]. The formation of this water film can be interpreted by following the theory of structure. To a certain extent, the “water film” could obstruct the crude oil from reabsorbing on the surfaces, accordingly improving the efficiency of residual oil displacement. Therefore, the PHDB-SiO_2_ nanoparticle composite displacement fluid flooding for low-permeability reservoirs demonstrates a great potential for improving oil recovery. The performance of this PHDB-SiO_2_ nanofluids flooding was compared to other displacement agents in Table 3 [2,5,7,10,19,23,38].

## 4. Conclusions

The recovery performance of residual oil is restrained given the characteristics of poor permeability, low porosity, heterogeneity, and the obvious capillary effect in low-permeability reservoirs. Therefore, methods to recover these residual oils efficiently are important to guarantee energy supply. A novel “microbial-nanofluid” composed of PHDB and SiO_2_ nanoparticles was constructed for enhancing residual oil recovery. The visual glass micromodel displacement experimental results showed that the optimum composite concentration of microbial-nanofluids were PHDB (7.0%, *v*/*v*) and SiO_2_ nanoparticles (100 mg/L), and the residual oil recovery was increased by 30.1%, 12.4%, and 8.5%, respectively, compared with water flooding SiO_2_ nanoparticles flooding, and PHDB flooding. The morphological characteristics of residual oil mobilization after PHDB-SiO_2_ nanoparticle composite flooding were mainly small and dispersed oil droplets in the excessive areas. Furthermore, clarified the synergistic mechanism of four kinds of PHDB-SiO_2_ nanoparticle composite flooding to enhance the residual oil recovery in low-permeability reservoirs. Based on the above results, the microbial-nanofluids flooding for low-permeability reservoirs demonstrates a great potential for improving oil recovery.

## Figures and Tables

**Figure 1 microorganisms-10-02104-f001:**
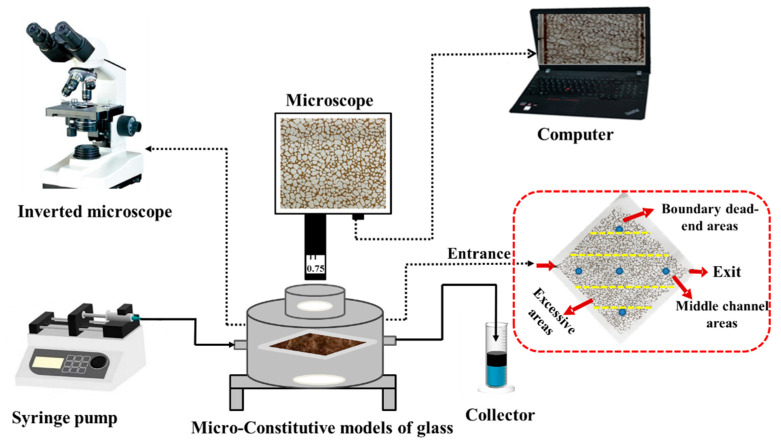
Schematic diagram of residual oil-displacement experimental equipment process by micromodels.

**Figure 2 microorganisms-10-02104-f002:**
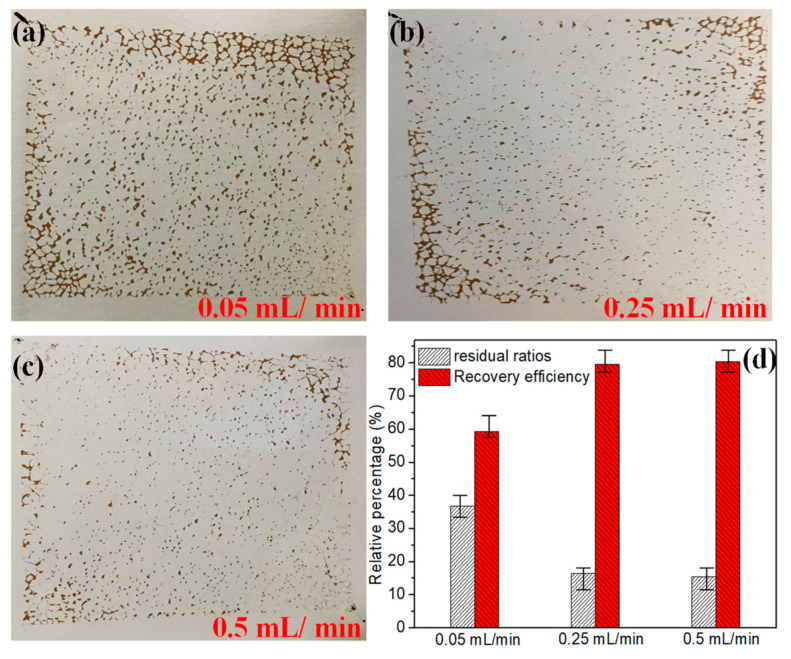
Distribution characteristics and recovery efficiency of residual oil under different water flooding rates. (**a**) 0.05, (**b**) 0.25, (**c**) 0.5, and (**d**) oil recovery efficiency.

**Figure 3 microorganisms-10-02104-f003:**
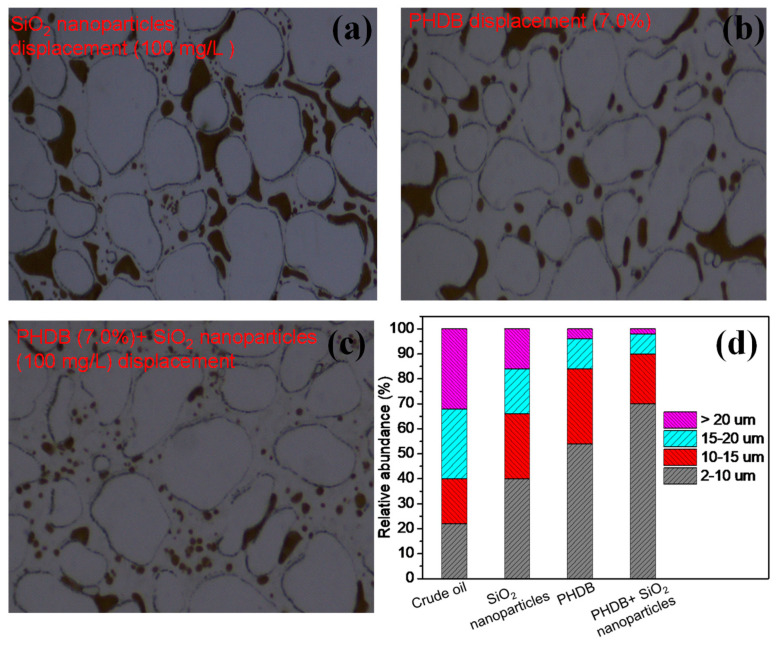
Characteristics of distribution and formation of residual oil under the maximum concentration of different displacement fluids (**a**–**c**), and corresponding droplet size distribution of residual oil emulsion (**d**).

**Figure 4 microorganisms-10-02104-f004:**
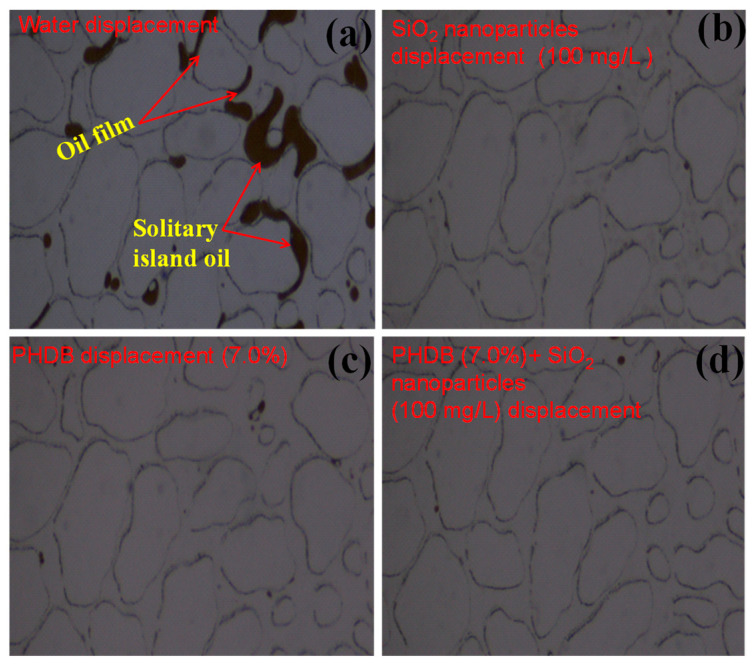
Mobilization characteristics of residual oil in middle channel areas under different displacement fluids. (**a**) After water flooding, (**b**) 100 mg/L of SiO_2_ nanoparticle displacement, (**c**) 7.0% of PHDB displacement, and (**d**) PHDB (7.0%)-SiO_2_ nanoparticles (100 mg/L) composite displacement.

**Figure 5 microorganisms-10-02104-f005:**
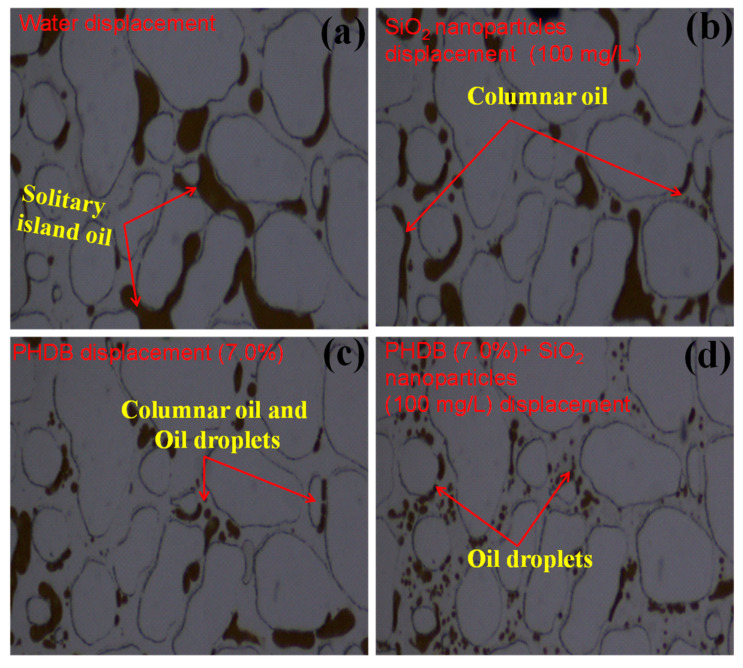
Mobilization characteristics of residual oil in excessive areas under different displacement fluids. (**a**) After water flooding, (**b**) 100 mg/L of SiO_2_ nanoparticle displacement, (**c**) 7.0% of PHDB displacement, and (**d**) PHDB (7.0%)-SiO_2_ nanoparticles (100 mg/L) composite displacement.

**Figure 6 microorganisms-10-02104-f006:**
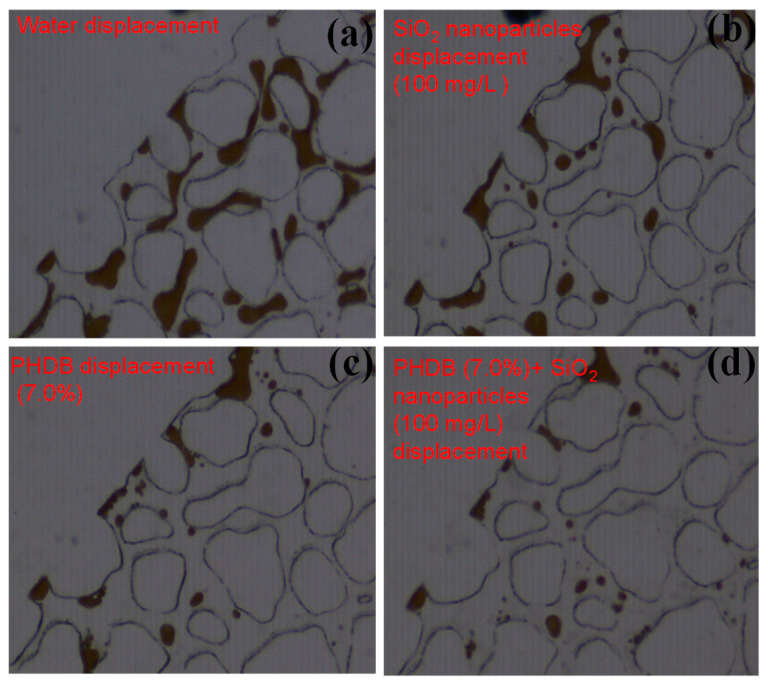
Mobilization characteristics of residual oil in boundary dead-end areas under different displacement fluids. (**a**) After water flooding, (**b**) 100 mg/L of SiO_2_ nanoparticle displacement, (**c**) 7.0% of PHDB displacement, and (**d**) PHDB (7.0%)-SiO_2_ nanoparticles (100 mg/L) composite displacement.

**Figure 7 microorganisms-10-02104-f007:**
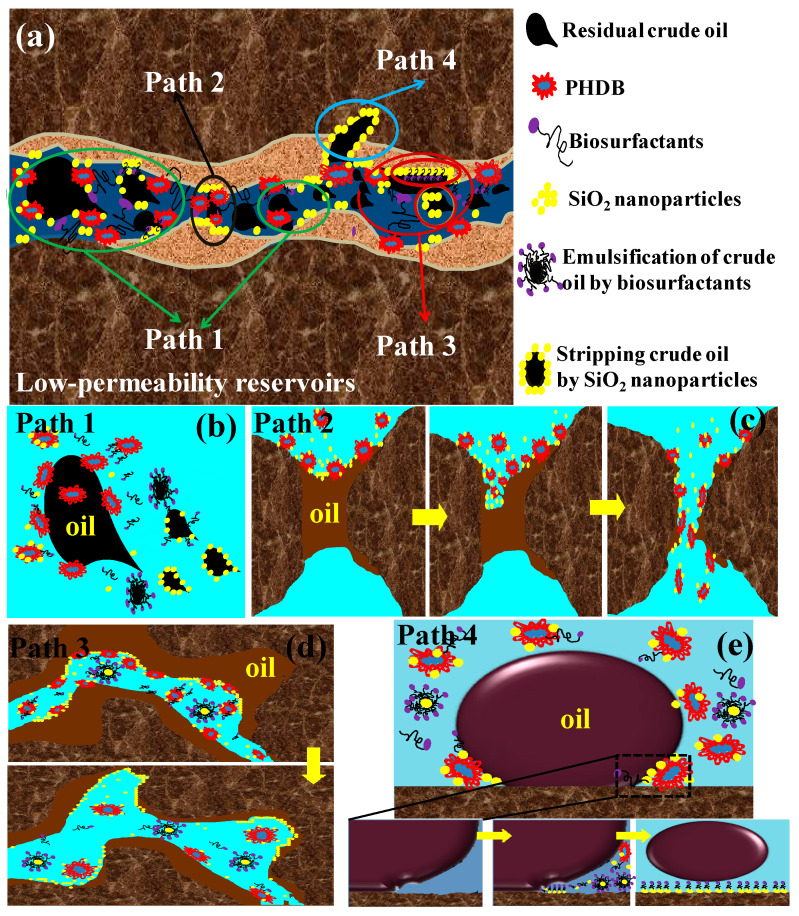
Schematic diagram of the mechanism of PHDB-SiO_2_ nanoparticle composite flooding for enhancing oil recovery. (**a**) Collaborative pathways for EOR by PHDB-SiO_2_ nanoparticle composite displacement, (**b**) Emulsification and dispersion of residual crude oil, (**c**) The clogging channels for the residual oil were dredged by PHDB-SiO_2_ nanoparticle composite displacement, (**d**) The residual crude in the dead-end pores were displaced by PHDB-SiO_2_ nanoparticle composite displacement, and (**e**) The residual crude oil being stripped and displaced from the rockface by PHDB-SiO_2_ nanoparticle composite displacement.According to Figure 2 and Appendix A, a large amount of residual oil was easily trapped in the pore-throat structure and resulted in the oil layer on the pore wall after the primary water flooding. The swept volume was small because the displacing fluid could only be in contact with one side of the oil in the middle channel pores [23,32]. However, with the effect of PHDB-SiO_2_ nanoparticle composite flooding, the residual oil ratio in the pore-throat decreased obviously, corresponding to an enlarged water sweep area (Appendix A). This is because PHDB can break the oil into small droplets, resulting in accelerating emulsification and greatly promoting the dispersion of residual oil (Figure 7b), thereby increasing the contact probability between SiO_2_ nanoparticles and oil droplets, and promoting the adsorption and emulsification of the residual oil by nanoparticles (Path 1). After that, the residual oil droplets encapsulated by the nanoparticles were more propitious to transport and result in EOR.

**Table 1 microorganisms-10-02104-t001:** Design of the micromodel displacement tests.

Experimental Number	Primary Flooding (0.05 mL/min)	Slug Displacement Fluids (3.0 PV)	Objectives
1	Water flooding	Produced water (sterilization)	Experimental control
2	Water flooding	PHDB fluids (3%, 5%, and 7%)	To screen the optimal PHDB concentration
3	Water flooding	SiO_2_ nanoparticles fluids (50, 100, and 200 mg/L)	To screen the optimal nanoparticles concentration
4	Water flooding	PHDB (7%) + SiO_2_ nanoparticles (100 mg/L) fluids	To investigate the EOR effects of composite displacement

**Table 2 microorganisms-10-02104-t002:** Residual oil recovery efficiency under different displacement fluids.

Displacement Fluids	Injection Concentration	Initial Residual Oil Ratio, %	Final Residual Oil Ratio, %	Average Recovery Efficiency, %
Water	—	70 ± 3	36.8 ± 2.5	68.8
SiO_2_ nanoparticles	50 mg/L	70 ± 3	7.3 ± 1.5	82.6
SiO_2_ nanoparticles	100 mg/L	70 ± 3	4.8 ± 0.5	86.5
SiO_2_ nanoparticles	200 mg/L	70 ± 3	6.1 ± 0.4	84.4
PHDB	3.0% *v*/*v*	70 ± 3	4.2 ± 0.3	87.8
PHDB	5.0% *v*/*v*	70 ± 3	3.8 ± 0.2	89.3
PHDB	7.0% *v*/*v*	70 ± 3	2.1 ± 0.3	90.5
PHDB + SiO_2_ nanoparticles	7.0% *v*/*v* + 100 mg/L	70 ± 3	1.0 ± 0.2	98.9

**Table 3 microorganisms-10-02104-t003:** Performance of different oil-displacement technologies on crude oil recovery in the low-permeability reservoir.

Displacement Agent	Injection Concentration	Primary Flooding Method	Injection Method	Final OR, %	Relative Water Flooding EOR, %	Ref.
Water	—	Waterflooding	Normal waterflooding	40–50%	—	[2]
Chemical agents	100 mg/L	Waterflooding	Slug displacement	60–70%	20–30%	[10]
SiO_2_ nanoparticles	100 mg/L	Waterflooding	Slug displacement	>80%	>20%	[5]
Biosurfactants	100 mg/L	Waterflooding	Slug displacement	70–80%	20–30%	[7]
SiO_2_ nanoparticles +Biosurfactants	50 mg/L +50 mg/L	Waterflooding	Slug displacement	>80%	>25%	[23]
Other microbial fluids	7.0% *v*/*v*	Waterflooding	Slug displacement	50–60%	10–20%	[38]
PHDB fluids	7.0% *v*/*v*	Waterflooding	Slug displacement	>70%	>20%	[19]
PHDB+SiO_2_ nanoparticles	7.0% *v*/*v* +100 mg/L	Waterflooding	Slug displacement	>90%	>30%	This work

## Data Availability

Not applicable.

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
