# Peer review of "New Technique for Enhancing Residual Oil Recovery from Low-Permeability Reservoirs: The Cooperation of Petroleum Hydrocarbon-Degrading Bacteria and SiO2 Nanoparticles"

_microorganisms, 2022, doi:10.3390/microorganisms10112104_

Round 1
Reviewer 1 Report
In the introduction, the authors mentioned that "Conventional recovery techniques are inefficient". Please add to the results a Table that permits comparing the different methods reported in the literature and compare with the results obtained in this research.
Table 2. Residual oil recovery efficiency. In my opinion, it is necessary to present any statistical data as average and Standart deviation. Why it is not reported?
Figure 3. The authors mentioned: "Characteristics of distribution and formation of residual oil under the optimal concentration of different displacement fluids". The word "Optimal" is not correct. The optimal value is always linked with an objective function, in this context, the concentration must be described as the maximum concentration.
In the conclusion section the authors indicate that "Based on the above results, the microbial-nanofluids flooding for low permeability reservoirs demonstrates a great potential for improving oil recovery". This phrase must be supported by the results compared with other works published in the literature. Please, add a comparison table describing the results of other techniques and compare with the data obtained in this research.
Author Response
Dear Reviewer:
On behalf of co-authors, I sincerely thank you for your very valuable comments on our manuscript entitled “New technique for enhancing residual oil recovery from low permeability reservoirs: The cooperation of petroleum hydro-carbon-degrading bacteria and SiO2 nanoparticles” (No. Microorganisms-1939242). We have revised the manuscript carefully according to your comments. All changes made to the manuscript were highlighted in red color. Attached with the revised version, we would like to resubmit it for your kind consideration.
Looking forward to hearing from you.
Thank you and best regards.
Prof. Kun Guo
Responses to Reviewer Comments
Q1: In the introduction, the authors mentioned that "Conventional recovery techniques are inefficient". Please add to the results a Table that permits comparing the different methods reported in the literature and compare with the results obtained in this research.
Comment: Thanks for giving us the suggestions and comments to improve the quality of the manuscript. “Table 3” has been added in Section 3.4 in the revised manuscript.
Action: Page 14, Line 487-494: “Table 3” has been added in Section 3.4, and “Therefore, the PHDB-SiO2 nanoparticles composite fluid flooding for low-permeability reservoirs demonstrated a great potential for improving oil recovery. The performance of this PHDB- SiO2 nanofluids flooding was compared to other displacement agents in Table 3 [2,5,7,10,19,39,40]” has been added in the revised manuscript.
Table 3. Performance of different oil displacement technologies on crude oil recovery in the low-permeability reservoir.
|
Displacement agent |
Injection concentration |
Primary flooding method |
Injection method |
Final OR, % |
Relative water flooding EOR, % |
Ref. |
|
Water |
— |
Waterflooding |
Normal waterflooding |
40-50% |
— |
2 |
|
Chemical agents |
100 mg/L |
Waterflooding |
Slug displacement |
60-70% |
20-30% |
10 |
|
SiO2 nanoparticles |
100 mg/L |
Waterflooding |
Slug displacement |
>80% |
>20% |
5 |
|
Biosurfactants |
100 mg/L |
Waterflooding |
Slug displacement |
70-80% |
20-30% |
7 |
|
SiO2 nanoparticles + Biosurfactants |
50 mg/L +50 mg/L |
Waterflooding |
Slug displacement |
>80% |
>25% |
39 |
|
Other microbial fluids |
7.0% v/v |
Waterflooding |
Slug displacement |
50-60% |
10-20% |
40 |
|
PHDB fluids |
7.0% v/v |
Waterflooding |
Slug displacement |
>70% |
>20% |
19 |
|
PHDB+SiO2 nanoparticles |
7.0% v/v +100 mg/L |
Waterflooding |
Slug displacement |
>90% |
>30% |
This work |
Q2: Table 2. Residual oil recovery efficiency. In my opinion, it is necessary to present any statistical data as average and Standard deviation. Why it is not reported?
Comment: Thanks for giving us the suggestions and comments to improve the quality of the manuscript. We calculated the average value and analyzed the standard deviation of the residual oil recovery efficiency data in Table 2.“Table 2” has been revised in the revised manuscript.
Action: Page 8, Line 287: “Table 2” has been revised in the revised manuscript.
Table 2. Residual oil recovery efficiency under different displacement fluids
|
Displacement fluids |
Injection concentration |
Initial residual oil ratio, % |
Final residual oil ratio, % |
Average recovery efficiency, % |
|
Water |
— |
70±3 |
36.8±2.5 |
68.8 |
|
SiO2 nanoparticles |
50 mg/L |
70±3 |
7.3±1.5 |
82.6 |
|
SiO2 nanoparticles |
100 mg/L |
70±3 |
4.8±0.5 |
86.5 |
|
SiO2 nanoparticles |
200 mg/L |
70±3 |
6.1±0.4 |
84.4 |
|
PHDB |
3.0% v/v |
70±3 |
4.2±0.3 |
87.8 |
|
PHDB |
5.0% v/v |
70±3 |
3.8±0.2 |
89.3 |
|
PHDB |
7.0% v/v |
70±3 |
2.1±0.3 |
90.5 |
|
PHDB+SiO2 nanoparticles |
7.0% v/v+100 mg/L |
70±3 |
1.0±0.2 |
98.9 |
Q3: Figure 3. The authors mentioned: "Characteristics of distribution and formation of residual oil under the optimal concentration of different displacement fluids". The word "Optimal" is not correct. The optimal value is always linked with an objective function, in this context, the concentration must be described as the maximum concentration.
Comment: Thank you for your constructive suggestion. In Figure 3, “optimal concentration” has been revised to “maximum concentration” in the revised manuscript.
Action: Page 9, Line 308: In Figure 3, “optimal concentration” has been revised to “maximum concentration” in the revised manuscript.
Q4: In the conclusion section the authors indicate that "Based on the above results, the microbial-nanofluids flooding for low permeability reservoirs demonstrates a great potential for improving oil recovery". This phrase must be supported by the results compared with other works published in the literature. Please, add a comparison table describing the results of other techniques and compare with the data obtained in this research.
Comment: Thanks for giving us the suggestions and comments to improve the quality of the manuscript. “Table 3” has been added in Section 3.4 in the revised manuscript.
Action: Page 14, Line 487-494: “Table 3” has been added in Section 3.4, and “Therefore, the PHDB-SiO2 nanoparticles composite fluid flooding for low-permeability reservoirs demonstrates a great potential for improving oil recovery. The performance of this PHDB- SiO2 nanofluids flooding was compared to other displacement agents in Table 3 [2,5,7,10,19,39,40]” has been added in the revised manuscript.
Table 3. Performance of different oil displacement technologies on crude oil recovery in the low-permeability reservoir.
|
Displacement agent |
Injection concentration |
Primary flooding method |
Injection method |
Final OR, % |
Relative water flooding EOR, % |
Ref. |
|
Water |
— |
Waterflooding |
Normal waterflooding |
40-50% |
— |
2 |
|
Chemical agents |
100 mg/L |
Waterflooding |
Slug displacement |
60-70% |
20-30% |
10 |
|
SiO2 nanoparticles |
100 mg/L |
Waterflooding |
Slug displacement |
>80% |
>20% |
5 |
|
Biosurfactants |
100 mg/L |
Waterflooding |
Slug displacement |
70-80% |
20-30% |
7 |
|
SiO2 nanoparticles + Biosurfactants |
50 mg/L +50 mg/L |
Waterflooding |
Slug displacement |
>80% |
>25% |
39 |
|
Other microbial fluids |
7.0% v/v |
Waterflooding |
Slug displacement |
50-60% |
10-20% |
40 |
|
PHDB fluids |
7.0% v/v |
Waterflooding |
Slug displacement |
>70% |
>20% |
19 |
|
PHDB+SiO2 nanoparticles |
7.0% v/v +100 mg/L |
Waterflooding |
Slug displacement |
>90% |
>30% |
This work |
Reviewer 2 Report
GENERAL COMMENTS:
This paper successfully presents a new technique using bacteria-SiO2 nanoparticles and validation of the effectiveness of the “microbial-nanofluids” to increase oil recovery. This study is both novel and interesting for the MDPI readership. However, there are a few issues (specific comments) that need to be addressed before it can go into the press.
SPECIFIC COMMENTS:
1. Introduction
The second paragraph is too long.
2. Materials and Methods
2.1. Materials
It was not clear how the PHDB strain was isolated from the production fluid of the Xinjiang oil field in China. Clarify.
2.2. Preparation of the displacement fluids
PHDB displacement fluids were prepared at concentrations of 3.0%, 5.0%, and 7.0%, v/v and SiO2 nanoparticles displacement fluids at concentrations of 50 mg/L, 100 mg/L, and 200 mg/L. What criteria were used to choose 7% and 100 mg/L to prepare the PHDB-SiO2 nanoparticles composite displacement fluid? This is only clear in the results.
2.4. Micromodel displacement experiment
In Table 1, the concentration of SiO2 nanoparticles (200 mg/L) for preparing the PHDB-SiO2 nanoparticles composite displacement fluid is different from what was shown in item 2.2 (100 mg/L). What concentration was used in the EOR experiment, 100 or 200 mg/L? Check and correct.
3.2. Effects of displacement fluids on residual oil recovery
In this item, the authors show that crude oil displacement is favored by the formation of emulsion by the joint action of the PHDB-SiO2 nanoparticle composite. Thinking about a production scale, what would it be like to produce this emulsified crude oil? The authors thought what would be the consequences of using PHDB-SiO2 nanoparticle composite to flow assurance? Could you comment on this?
3.4. Theoretical mechanism analysis of microbial-nanofluids for EOR
In this item, the authors comment that “the PHDB degrades n-alkanes and cycloalkanes with different carbon numbers without other carbon sources”. In this case, what are the characteristics of the crude oil produced? Would the use of the PHDB-SiO2 nanoparticle composite produce crude oil with more heavy compounds and, therefore, of lower economic value? In this case, what would be the advantages of using the PHDB-SiO2 nanoparticle composite for the petroleum industry? It would be valid to test the PHDB-SiO2 nanoparticle composite with low API crude oils to evaluate its efficiency against heavy oils.

Author Response
Dear Reviewer:
On behalf of co-authors, I sincerely thank you for your very valuable comments on our manuscript entitled “New technique for enhancing residual oil recovery from low permeability reservoirs: The cooperation of petroleum hydro-carbon-degrading bacteria and SiO2 nanoparticles” (No. Microorganisms-1939242). We have revised the manuscript carefully according to your comments. All changes made to the manuscript were highlighted in red color. Attached with the revised version, we would like to resubmit it for your kind consideration.
Looking forward to hearing from you.
Thank you and best regards.
Prof. Kun Guo
Responses to Reviewer Comments
Q1: Introduction: The second paragraph is too long.
Comment: Thanks for giving us the suggestions and comments to improve the quality of the manuscript. We have made a major revision to the second paragraph in Introduction Section. All the changes were highlighted in red color.
Action: Page 2, Line 58-78: The second paragraph in Introduction Section has been revised in the revised manuscript. Revised the following:
Compared with traditional tertiary oil recovery technologies, nanofluids as a new type of oil-displacement agent have become a hot topic in recent years to enhance oil recovery in low permeability reservoirs, where silica (SiO2)-based nanoparticles have been most commonly used [6,7]. The nanoparticles exhibit marvelous interfacial behavior due to their na-noscale size and large specific surface area [8]. Hendraningrat et al utilized hydrophilic SiO2 nano-particles to prepare nanofluids and studied the effect of nanofluid flooding on enhanced oil recovery of low-medium permeability sandstone [9]. Besides, the co-adsorption of nanoparticles and surfactants on oil/water or oil/water/solid can affect the interface energy, resulting in interfacial tension reduction and wettability alteration [10,11]. Nwidee et al revealed that nanoparticle-biosurfactant systems exhibited an excellent ability as a new EOR agent under reservoir conditions [12].
Nanofluid is defined as a stable and homogeneous suspension obtained by adding nanoparticles with 1-100 nm average size in a traditional liquid medium such as water, oil phase, or alcohol [10]. Even though nanofluids have shown great potential in the EOR process, the nano-suspension prepared by pure nanoparticles or nanoparticles and biosurfactants cannot obtain effective production. Meanwhile, these nanofluid products show good oil displacement performance in the laboratory environment, but they often have agglomeration, dissolution, and instability defects in the real reservoir environment [13], which limit their oil displacement ability. For example, surfactants will be adsorbed and retained in the reservoir, polymers will degrade under high temperatures and high salinity, and alkali flooding and combination flooding will corrode pipelines in the process of oil production and scale is easy to appear in the reservoir [10,11]. Based on these challenges, it is necessary to modify the action environment of SiO2 nanoparticles to achieve the flooding effect of the composite system.
Q2: In Section 2.1, It was not clear how the PHDB strain was isolated from the production fluid of the Xinjiang oil field in China. Clarify.
Comment: Thank you for your constructive suggestion. We have added the PHDB strain isolation process in Section 2.1.
Action: Page 3-4, Line 132-139: “The experimental process of the PHDB isolation is as follows: 10 mL of the produced water and 90 mL of PHDB screening medium (molasses 1.47 g/L, NaNO3 5.33 g/L, (NH4)2SO4 2.67 g/L, NaCl 2.0 g/L, KH2PO4 10.0 g/L, Na2HPO4 4.0 g/L, MgSO4 0.5 g/L, and yeast powder 0.6 g/L)) were mixed at 45°C with shaking at 150 r/min for 3-5 days. Then, 5% of the mixed bacteria liquid was inoculated into 100 mL of fresh screening medium. The tubes were incubated for 3-5 days under facultative anaerobic conditions at 45°C with shaking at 150 r/min. Repeat the above steps several times, and it was then stretched on the petri dish to pick a single PHDB colony” has been added in the revised manuscript.
Q3: In Section 2.2, PHDB displacement fluids were prepared at concentrations of 3.0%, 5.0%, and 7.0%, v/v and SiO2 nanoparticles displacement fluids at concentrations of 50 mg/L, 100 mg/L, and 200 mg/L. What criteria were used to choose 7% and 100 mg/L to prepare the PHDB-SiO2 nanoparticles composite displacement fluid? This is only clear in the results.
Comment: Thanks for giving us the suggestions and comments to improve the quality of the manuscript. The standards for preparation of PHDB (7%, v/v)-SiO2 nanoparticles (100 mg/L) composite displacement fluid are based on the results of previous laboratory studies [5,7,19].
Action: Page 4, Line 155-157: “Based on the results of previous laboratory studies [5,7,19], the preparation standards of PHDB-SiO2 nanoparticles composite displacement fluid was designed.” has been added in the revised manuscript.
Q4: In Section 2.4, Table 1, the concentration of SiO2 nanoparticles (200 mg/L) for preparing the PHDB-SiO2 nanoparticles composite displacement fluid is different from what was shown in item 2.2 (100 mg/L). What concentration was used in the EOR experiment, 100 or 200 mg/L? Check and correct.
Comment: We apologize for our negligence of statements in the manuscript. Table 1, the SiO2 nanoparticles (100 mg/L) were added to the PHDB displacement fluid to prepare the PHDB-SiO2 nanoparticles composite displacement fluid. “200 mg/L” was an error writing, it has been revised to “100 mg/L” in the revised manuscript.
Action: Page 6, Line 217: Table 1, “200 mg/L” has been revised to “100 mg/L” in the revised manuscript.
Q5: In Section 3.2, In this item, the authors show that crude oil displacement is favored by the formation of emulsion by the joint action of the PHDB-SiO2 nanoparticle composite. Thinking about a production scale, what would it be like to produce this emulsified crude oil? The authors thought what would be the consequences of using PHDB-SiO2 nanoparticle composite to flow assurance? Could you comment on this?
Comment: Thank you for your constructive suggestion. In this study, the mixture of PHDB and SiO2 nanoparticles were used to prepare the composite displacement fluid was based on micromodel experiment. Since the microbial environment conditions cannot be simulated for the laboratory experiment, the micromodel experiment was designed to be mixed with SiO2 nanomaterials after the exogenous PHDB activated and cultured. Considering the production scale of this displacement fluid, because of the large number of PHDB in the real reservoir, the industrial PHDB-SiO2 nanoparticles composite displacement fluid is mainly microbial nutrient activators and SiO2 nanomaterials. Thus, the injection and flow of the composite displacement fluid in the reservoir are very good. The nutrient activates PHDB in situ in the reservoir while coupling with the SiO2 nanoparticles to the emulsification and displacement of residual crude oil.
Action: Page 9, Line 343-347: “Additionally, considering the abundance of PHDB in the real reservoir environment, the industrial PHDB-SiO2 nanoparticles composite displacement fluid are mainly microbial nutrient activators and SiO2 nanomaterials. The nutrient activates PHDB in situ in the reservoir while coupling with the SiO2 nanoparticles to emulsify and displace the residual crude oil, and to EOR” has been added in the revised manuscript.
Q6: In Section 3.4, In this item, the authors comment that “the PHDB degrades n-alkanes and cycloalkanes with different carbon numbers without other carbon sources”. In this case, what are the characteristics of the crude oil produced? Would the use of the PHDB-SiO2 nanoparticle composite produce crude oil with more heavy compounds and, therefore, of lower economic value? In this case, what would be the advantages of using the PHDB-SiO2 nanoparticle composite for the petroleum industry? It would be valid to test the PHDB-SiO2 nanoparticle composite with low API crude oils to evaluate its efficiency against heavy oils.
Comment: Thank you for your constructive suggestion. In PHDB-SiO2 nanoparticle composite displacement technology, PHDB can use crude oil as a carbon source for growth and corresponding metabolic activities (such as producing biosurfactants and bio-acids). However, PHDB growth consumes very little crude oil relative to reservoir reserves, so it will not affect the quality of crude oil. On the contrary, according to the lab's previous research [2], PHDB can degrade the heavy components of crude oil (such as aromatic compounds (phenanthrenes, benzenes, naphthalenes, and thiophenes)) and make it lighter to improve the quality of ten oil.
Additionally, the PHDB-SiO2 nanoparticle composite flooding does not produce more heavy compounds in the crude oil, because most of the PHDB-SiO2 nanoparticle composite displacement fluids remain in the reservoir, and the produced oil settling pool, and the mixture of crude oil, PHDB, and SiO2 nanoparticles in settling pool is re-injected into the reservoir to dislodge the crude oil. The advantage of using the PHDB-SiO2 nanoparticle composite for the petroleum industry is a green, biodegradable oil displacement material with low post-processing costs compared to chemical oil displacement agents. The viscosity and density of crude oil produced by PHDB-SiO2 nanoparticle composite (0.883 g/cm3 and 18.5 mPa·s at 30 °C) is similar to the real crude oil (0.879 g/cm3 and 18.2 mPa·s at 30 °C), and the test oil is simulated light crude oil because of the micro model experiment. Based on the consideration of the reviewers, in later experimental studies, we will carry out to test of the PHDB-SiO2 nanoparticle composite with low API crude oils to evaluate its efficiency against heavy oils.
Action: Page 3, Line 94-99: “It's worth noting that the PHDB growth consumes very little crude oil relative to reservoir reserves, so it will not affect the quality of crude oil. On the contrary, according to the lab's previous research [2], PHDB can degrade the heavy components of crude oil (such as aromatic compounds (phenanthrenes, benzenes, naphthalenes, and thiophenes)) and make it lighter to improve the quality of oil” has been added in the revised manuscript.

Round 2
Reviewer 1 Report
The authors have made all suggested changes